# Structural basis of lipid-droplet localization of 17-beta-hydroxysteroid dehydrogenase 13

Shenping Liu [1] ✉, Ruth F. Sommese[1] ✉, Nicole L. Nedoma [1],
Lucy Mae Stevens[1], Jason K. Dutra[2], Liying Zhang[2,4], David J. Edmonds [2,5],
Yang Wang[2], Michelle Garnsey [2] & Michelle F. Clasquin [3]

Hydroxysteroid 17-beta-dehydrogenase 13 (HSD17B13) is a hepatic lipid droplet-associated enzyme that is upregulated in patients with non-alcoholic fatty liver disease. Recently, there have been several reports that predicted loss of function variants in HSD17B13 protect against the progression of steatosis to non-alcoholic steatohepatitis with fibrosis and hepatocellular carcinoma. Here we report crystal structures of full length HSD17B13 in complex with its NAD$^+$ cofactor, and with lipid/detergent molecules and small molecule inhibitors from two distinct series in the ligand binding pocket. These structures provide insights into a mechanism for lipid droplet-associated proteins anchoring to membranes as well as a basis for HSD17B13 variants disrupting function. Two series of inhibitors interact with the active site residues and the bound cofactor similarly, yet they occupy different paths leading to the active site. These structures provide ideas for structure-based design of inhibitors that may be used in the treatment of liver disease.

Hydroxysteroid dehydrogenases (HSD) comprise a large family of enzymes that are involved in the biogenesis and metabolism of many steroid and non-steroid substrates in animals and human[1]. They are nicotinamide adenine dinucleotide (phosphate) (NAD$^+$(P)/NAD(P)H)-dependent oxidoreductases, which interconvert ketones and the corresponding secondary alcohols at different positions of steroidal substrates (3α-, 3β-, 11β-, 17β-, 20α-, and 20β-position)[2]. Given their roles in steroid metabolism, HSD family members play important biological roles in human health and have been associated with a variety of diseases[1,3–7]. For example, members of the 17β-HSD subfamily have been associated with breast and prostate cancer, polycystic kidney cancer, and Alzheimer's disease, to name a few[8].

Recently, one of those members, 17β-HSD13 (HSD17B13), has garnered significant interest after a potential role in the pathogenesis of non-alcoholic fatty liver disease (NAFLD) was identified through genome-wide association studies[9–13]. NAFLD is a serious disease that affects around one-quarter of the global population[14]. The first breakthrough towards identifying the role of HSD17B13 in NAFLD pathogenesis in humans occurred when an exome-wide association study revealed that a splice variant, rs72613567 or isoform D (isoD), conferred a protective effect against NAFLD disease severity. This variant was strongly associated with slowed progression of NAFLD toward non-alcoholic steatohepatitis (NASH) and advanced fibrosis[10]. The isoD variant is the product of insertion of adenine adjacent to the donor splice site of exon 6 (T > TA), which disrupts mRNA splicing. The isoD variant was shown to be translated into a truncated protein (P274del) that was unstable, with low protein level in human liver biopsy samples and decreased enzymatic activity against β-estradiol in vitro and in cell-based overexpression assays[10].

Subsequent genome-wide association studies have identified several additional human HSD17B13 variants that are protective against progression to advanced NASH and liver injury. These variants either strongly shifted the splicing equilibrium toward isoD[11–13], resulted in a truncated protein product distinct from IsoD (Ala192LeuFsTERM or A192Lfs)[12], or resulted in non-synonymous mutations (P260S)[11]. A192Lfs and P260S proteins are predicted to be unstable, with decreased protein expression levels, and indeed overexpressed P260S variant protein lacked enzyme activity. This together with the fact that

[1]Medicine Design, Pfizer Inc, Groton, CT 06340, USA. [2]Medicine Design, Pfizer Inc, Cambridge, MA 02139, USA. [3]Internal Medicine Research Unit, Pfizer Inc, Cambridge, MA 02139, USA. [4]Present address: Discovery Chemistry, Merck Research Laboratories, Cambridge, MA, USA. [5]Present address: Medicinal Chemistry, Roche, Basel, Switzerland. ✉e-mail: Shenping.Liu@pfizer.com; Ruth.Sommese@pfizer.com

HSD17B13 is significantly upregulated in the liver of patients with NASH[11,15], suggests it may be a therapeutic target for NASH. Whether HSD17B13 variants identified to date, therefore, protect against NASH in humans through loss of the scaffolding function or the enzymatic activity or both is unclear.

In fact, very little is known about the function of HSD17B13 beyond it being a liver-specific, lipid droplet (LD)-associated protein[16–18]. LDs are storage organelles at the center of lipid and energy homeostasis. They have a unique architecture consisting of a hydrophobic core of neutral lipids, which is enclosed by a phospholipid monolayer that is decorated by a specific set of proteins[19–22]. Although several domains of HSD17B13 were reported to be important for LD localization by mutagenesis studies[17,23–25], currently a structural understanding of HSD17B13 LD association is lacking. In fact, although up to 100–150 major proteins were identified as LD-associated proteins in mammalian cells[21,26], the mechanism of LD targeting of these proteins is still unclear[27].

Because loss of function HSD17B13 variants are protective against NASH, small molecule HSD17B13 inhibitors may serve as oral medications for liver diseases. A key gap for designing potent small molecule inhibitors to HSD17B13, however, is the lack of any published HSD17B13 crystal structures. The available structure of the closest member of 17β-HSD subfamily is that of human 17β-HSD11 (HSD17B11) (PDB ID 1YB1). Like HSD17B13, HSD17B11 is an LD-associated protein, and HSD17B13 and HSD17B11 share 67% and 79% identity and similarity in sequences, respectively, in their core catalytic domains. The available structure for HSD17B11, however, is truncated which was likely done to facilitate crystallization. Specifically, it is missing the initial highly hydrophobic, LD-targeting N-terminal 22 residues[28] and the final amphipathic C-terminal 25 residues, resulting in a soluble, non-membrane-associated form of the protein. Additionally, the structure lacks the NAD$^+$/NADH cofactor, and based on homology modeling, the active site is collapsed, and the protein is non-functional.

Here we report high-resolution crystal structures of cofactor and inhibitor-bound dog and human HSD17B13 proteins. These structures provide significant insight into the LD association of HSD17B13 (and HSD17B11) as well as the structural hypothesis behind the reported disease-associated variants. Combined with structure-based mutagenesis they further provide insight into substrate selectivity and inhibitor binding. Finally, comparing the structures of HSD17B13 with two small molecule inhibitors with distinct scaffolds highlights different potential vectors for structure-based design.

## Results

### Small molecule inhibitors of HSD17B13

To identify small molecules inhibitors of HSD17B13 we carried out a high-throughput biochemical screen against a Pfizer proprietary library of around 3.2 million singleton compounds at 10 μM final concentration, using purified human HSD17B13, NAD$^+$ cofactor, and β-estradiol as substrate. The screening gave a 1.8% hit rate (>45% inhibition at 10 μM). The hit list was then run in single point confirmation (10 μM) against HSD17B13 and a technology counter-screen that detected compounds that interfered with reaction product readout. Confirmed hits (>45% inhibition in HSD17B13 assay and clean in the counter-screen) were then run in IC$_{50}$ mode with HSD17B13. Two compounds with distinct scaffolds emerged as validated hits (Fig. 1 and Supplementary Figure 1). Compound **1** is a fluorophenol-containing compound that is a disclosed antagonist of N-methyl-D-aspartate (NMDA) NR2B receptor[29]. Compound **2** is a benzoic acid compound containing a sulfonamide linker, and it was reported to be an AKT1 kinase inhibitor[30]. Both were reasonably potent against HSD17B13 in biochemical assays using both β-estradiol or Leukotriene B4 (LTB4) as substrate and NAD$^+$ as cofactor. When tested in cell assay, **1** was found to be active and **2** inactive (Supplementary Figure 1). In our experience, it is not surprising that **2** is inactive in cells because it has two highly polar groups (sulfonamide and sulfone) that lead to low cell penetration.

### Structure determination of HSD17B13

HSD17B13 is an LD-associated protein[16], and at the N-terminus of the protein, there is a stretch of highly hydrophobic Leu and Ile residues (A1-15: MNIILEILLLLITII). Extensive protein engineering was pursued to remove these hydrophobic residues to favor protein purification and crystallization, but N-terminal truncation efforts generally resulted in low protein yield and/or aggregation (Supplementary Table 1). Two protective variants, IsoD and P260, were attempted and demonstrated reduced protein expression levels. In parallel, we also screened different expression systems (bacteria, Sf9, Expi293), various detergents, together with various engineered mutants, a few of which are listed in Supplementary Table 1. We eventually succeeded in generating and crystallizing the full-length human HSD17B13 protein with a C-terminal GSG linker and His tag to facilitate purification. The protein was extracted and solubilized using detergent micelles and octaethylene glycol monododecyl ether (C12E8), a non-ionic detergent, was identified as the suitable detergent that provided HSD17B13 with the largest stability needed for crystallization. The inclusion of cofactor NAD$^+$ and inhibitors were also critical for crystallization.

Initial crystallization of the human HSD17B13/compound **1** complex resulted in low-resolution (5–6 Å) diffracting crystals. Screening of HSD17B13 from different species for crystallization led to the much-improved crystals of dog wild-type HSD17B13 in complex with compound **1** and the cofactor, NAD$^+$ (Table 1, compound **1**-Dog WT). Subsequent mutation of four residues (G177, V178, T205, and I293) identified at the compound **1** binding site of dog HSD17B13 to the human ones (E177, G178, A205, and V293, dog mutant) further improved the resolution of these crystals (Table 1, compound **1**-Dog mutant). We could remove compound **1** from these crystals by soaking

Compound **1**
3-fluoro-N-({trans-4-[(2-fluorophenoxy)methyl]-1-hydroxycyclohexyl}methyl)-4-hydroxybenzamide
IC$_{50}$ = 0.18 ± 0.05 μM (n=9, β-estradiol as substrate)
IC$_{50}$ = 0.25 ± 0.09 μM (n=3, Leukotriene B4 as substrate)

Compound **2**
4-((2,5-dimethyl-3-tosylphenyl)sulfonamido)benzoic acid
IC$_{50}$ = 0.38 ± 0.25 μM (n=4, β-estradiol as substrate)
IC$_{50}$ = 0.45 ± 0.21 μM (n=3, Leukotriene B4 as substrate)

**Fig. 1 | Chemical structures of small molecule inhibitors studied.** The biochemical IC$_{50}$ values of inhibition against the β-estradiol and Leukotriene B4 as substrates oxidization by NAD$^+$ cofactor are listed, with number of experiments in parenthesis, each experiment was carried out with replicated measurements.

**Table 1 | Data collection and refinement statistics**

| Crystal[a] | Apo, Dog WT | Compound 1-Dog WT | Compound 1-Dog mutant | Compound 2-Human mutant |
|---|---|---|---|---|
| **Data collection** | | | | |
| Space group | P2$_1$2$_1$2 | P2$_1$2$_1$2 | P2$_1$2$_1$2 | P2$_1$ |
| Cells | | | | |
| a, b, c (Å) | 76.38 187.21 65.37 | 77.03 186.46 65.47 | 76.73 186.32 65.32 | 96.15 161.56 100.77 |
| α, β, γ (°) | 90, 90, 90 | 90, 90, 90 | 90, 90, 90 | 90 95.048 90 |
| Resolution (Å)[b] | 93.60–2.47 (2.64–2.47) | 71.19–2.23 (2.47–2.23) | 93.16–1.93 (2.21–1.926) | 100.38–2.65 (2.90–2.65) |
| Uni. reflections | 29,109 (1456) | 32,599 | 36,866 (1835) | 67,940 (3398) |
| R$_{pim}$[c] | 0.077 (0.579) | 0.039 (0.57) | 0.065 (0.404) | 0.083 (0.589) |
| I / σI | 8.3 (1.4) | 14.2(1.4) | 8.8 (1.8) | 7.9 (1.5) |
| Completeness (%) | 93.7 (50.6) | 92.9(58.7) | 93.6 (66.7) | 93.4 (56.3) |
| Redundancy | 6.6 (6.8) | 6.6 (6.6) | 6.2 (3.8) | 3.4 (3.4) |
| **Refinement** | | | | |
| Resolution (Å) | 30.86–2.47 | 23.16–2.22 | 21.1–1.91 | 37.9–2.65 |
| No. reflections | 29,084 (1449) | 32,551 (1607) | 36,628 (1441) | 67,909 (3399) |
| R$_{work}$[d] | 0.200 (0.265) | 0.210 (0.286) | 0.200 (0.223) | 0.212 (0.291) |
| R$_{free}$[e] | 0.227 (0.37) | 0.228 (0.251) | 0.224 (0.242) | 0.244 (0.338) |
| No. atoms | | | | |
| Protein | 4678 | 4553 | 4142 | 17,331 |
| Heterogen | 88 | 116 | 144 | 777 |
| Water | 136 | 81 | 73 | 23 |
| **B**–factors (Å$^2$) | | | | |
| Protein | 57.9 | 67.0 | 43.3 | 63.2 |
| Heterogen | 43.5 | 72.8 | 35.2 | 62.5 |
| Water | 48.4 | 46.6 | 24.3 | 35.6 |
| R.m.s. deviations | | | | |
| Bond lengths (Å) | 0.01 | 0.01 | 0.01 | 0.008 |
| Bond angles (°) | 1.1 | 1.1 | 1.1 | 0.98 |
| Ramachandran Favored, outlier (%)[f] | 97.8, 0.0 | 96.5, 0.3 | 96.9, 0.0 | 96.3, 0.5 |

[a]Data sets were collected on one crystal for each complex. Values in parentheses are for the highest-resolution shell.

[b]Statistics in the highest-resolution bins are shown in parenthesis. Based on anisotropic scaling and merge.

[c]$R_{pim} = \sum_{hkl} \sqrt[2]{1/(n-1)} \sum_i |I_i(hkl) - \langle I(hkl) \rangle| / \sum_{hkl} \sum_i I_i(hkl)$, where $I_i(hkl)$ is the $i$th intensity measurement of reflection $(hkl)$, $\langle I(hkl) \rangle$ is the mean intensity from multiple observations of that reflection, and $n$ is the number of observations of that reflection.

[d]$R_{work} = \sum_{hkl} ||F_{obs}| - |F_{calc}|| / \sum_{hkl} |F_{obs}|$, where $F_{obs}$ and $F_{calc}$ are observed and calculated structure-factor amplitudes, respectively.

[e]$R_{free}$ is calculated using 5% of reflections randomly excluded from refinement. For all crystals used in this study, the same set of reflections were chosen for exclusion.

[f]Percentages of residues in the most favorable and disallowed regions of the Ramachandran plot.

with buffer containing NAD$^+$ only and obtained NAD$^+$ bound dog HSD17B13 crystals (Table 1, apo Dog WT or apo hereafter). On the other hand, we could not generate compound **2** bound dog HSD17B13 crystals by soaking or co-crystallization and instead relied on co-crystallization with human HSD17B13. Based on findings from dog HSD17B13 crystals, we changed four surface residues distant from the active site of human HSD17B13 to those of dog protein at the dog HSD17B13 crystal interface (Q60K, I62R, R71H, and E161K, human mutant) to improve the resolution of the human HSD17B13 crystals and obtained useful crystals of the human mutant HSD17B13 in complex with NAD$^+$ and compound **2** (Table 1, compound **2**-human). The structures were solved using molecular replacement method using published HSD17B11 crystal structure (before the highly accurate protein structure program AlphaFold 2 was available[31]). There was one HSD17B13 dimer in the apo, compound **1**-Dog WT and compound **1**-Dog mutant crystals, and four HSD17B13 dimers in compound **2**-human mutant crystals. One of the possible reasons that there are four human HSD17B13 dimers in the crystal is that the ligand binding sites

contain lipids molecules (see below) which are heterogeneous in nature, making these complexes not structurally identical.

**Structural basis for the LD localization of HSD17B13**

In crystals both the human and the dog HSD17B13 formed dimers (Fig. 2). The overall structure of each HSD17B13 subunit could be divided into two parts: (i) the catalytic core domain (residues 29–259) that contained the cofactor binding site and the catalytic center and (ii) the membrane anchoring domain that consisted of the N-terminal helix (N2-P28) which was highly hydrophobic and a C-terminal proximal helix-turn-helix motif (P260-N286) which formed an amphipathic patch on the protein surface. Three proline residues at key positions in the protein marked the boundaries of these peptides: P28, P260, and P274. These prolines terminated the prior secondary structures and introduced sharp turns in the following peptides. Prolines are known to constrain the conformational space available to the protein backbone[32]. Interesting to note that both the P274del and P260S human variants were reported to be likely loss of function with lower protein stability and expression levels than the wild-type HSD17B13[10,11]. And consistently, modifications around P28 resulted in minimal protein expressions (Supplementary Table 1). The hydrophobic residues in the N-terminal helix of HSD17B13 were identified as critical for LD association in previous mutagenesis studies[23]. A similar N-terminal sequence in HSD17B11 was also shown to be essential for its LD localization[28].

The existence of the N-terminal hydrophobic helices and the amphipathic patches in the HSD17B13 dimer prompted us to model the interactions of the apo HSD17B13 dimer with lipid membranes (Fig. 2). In this model, the shape of HSD17B13 dimer resembles a sledge, with the N-terminal helices and the amphipathic patches running parallel to the surface of the lipid membrane. The N-terminal helices are at the lowest position of the HSD17B13 dimer facing down towards the lipid membrane, the helix-turn-helix motif at the second layer, and above them the catalytic cores of the HSD17B13 dimer. Based on this model, at the lowest position, the N-terminal hydrophobic helices will be immersed completely in the non-polar interior of a single leaflet of the lipid membrane, much like runners on a sledge. At the second level, the two "faces" of the amphipathic helix enable the hydrophobic face to enter the interior of the membrane, while the positively charged face interacts favorably with the negatively charged head groups of the membrane. The rigidity of helix-breaking P28 maintains a sharp angle between the N-terminal membrane anchoring helix and the following peptides leading to the catalytic core, ensuring that they cannot form transmembrane helices that cross a lipid bilayer. Consistent with HSD17B13's association with lipid droplets which are known to be formed by lipid monolayers surrounding cores of neutral lipids[16,19,20], the N-terminal helix lacks any positively charged residues which might otherwise mediate interactions with the lipid head groups of the second leaflet of a membrane bilayer. Based on our model, during the LD biogenesis, which was thought to be initiated by the accumulation of neutral lipids between the two leaflets of the endoplasmic reticulum (ER)[20], HSD17B13 would preferentially partition onto the monolayer of the budding LDs.

While the HSD17B13 dimer has structurally conserved catalytic cores, its N-terminal LD-targeting helices are unique when compared to four representative HSD proteins targeting different cellular organelles: HSD17B4, a soluble, cytosolic enzyme[33]; HSD17B1, a protein at equilibrium between the cytosol and membrane[34]; HSD11B1, an HSD with an N-terminal transmembrane helix targeting the lumen of ER[35]; and HSD17B11 (Supplementary Figure 2). When superimposed, the root-mean-square deviation (RMSD) was 1.2 Å for 258 Cα atoms of the core domains of dimers of HSD17B13 and HSD17B4 (PDB ID 1ZBQ). However, HSD17B4 lacks the N-terminal helices and the helix-turn-helix motifs which mediate HSD17B13's LD association. The structure of HSD17B1 dimer (PDB ID 1QYW) could be superimposed to that of

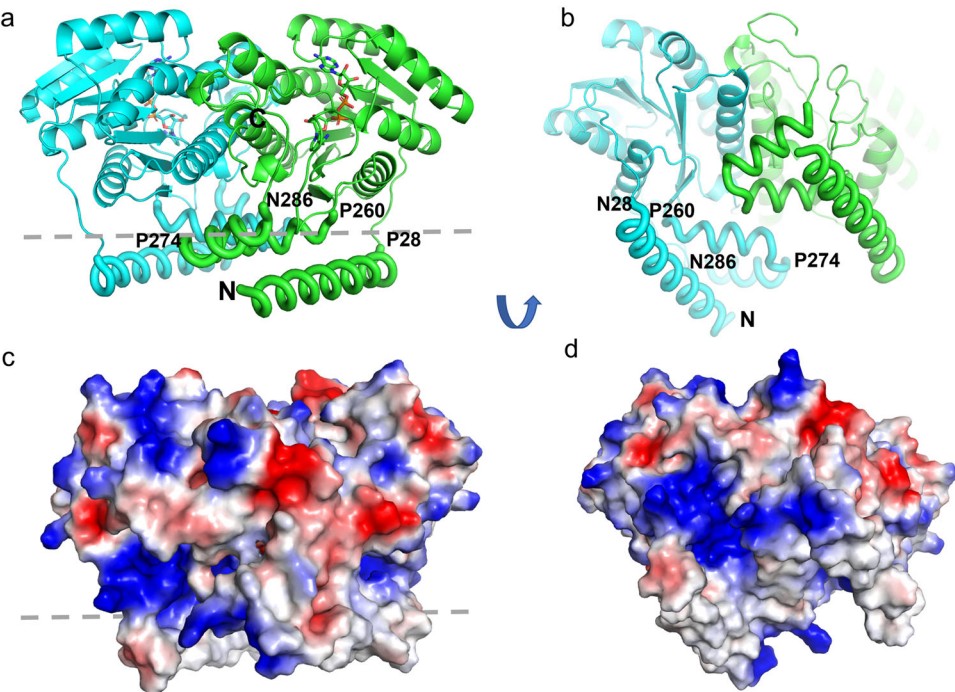

**Fig. 2 | Overall structure of dog apo HSD17B13. a, b** Ribbon diagrams of dog apo HSD17B13 dimer, with subunits colored green and cyan. The HSD17B13 dimer model was oriented with the putative membrane anchoring domains (tubes) down toward a phospholipid membrane. A gray dashed line is drawn to show the position of the membrane surface. The N- and C- termini as well as residues marking the boundaries of the membrane anchoring N-helix and helix-turn-helix are labeled for one subunit. **b** To highlight the membrane associating domains of the HSD17B13 dimer, the view is slightly tilted upwards relative to **a. c, d** surface electrostatic presentations of HSD17B13 dimer. Blue, red, and white represent positive, negative, and neutral surfaces, respectively. The view in **c** is the same as in **a**. The view in **d** is the same as in **b**.

HSD17B13 (RMSD 1.7 Å for 291 Cα). But while HSD17B1 has a similar amphipathic helix-turn-helix motif, it lacks the N-terminal membrane anchoring helices. The HSD11B1 dimer (PDB ID 1XSE) has similar catalytic cores (RMSD 2.00 Å for 350 Cα) and helix-turn-helix motifs. It also has the N-terminal membrane anchoring helices, but at their N-termini there are basic residues (K4/K5), critical for interacting the ER bilayer[36], which are absent in HSD17B13. Although the construct used in the published HSD17B11 structure (PDB ID 1YB1) lacked both the N-terminal helices and the helix-turn-helix motifs, the catalytic cores of HSD17B13 and HSD17B11 are similar (RMSD 0.6Å for 348 Cα).

Another key feature from the structure, is that the dimer interface of HSD17B13 is extensive (Supplementary Figure 3). Analysis of the dimer interface using program PISA[37] showed that the HSD17B13 dimer interface buried 2122 Å² surface area of each monomer. The main dimer interface involved residues 97 and 101, and peptides 128–157 and 175–207 in the catalytic core, burying 1736 Å² surface area. Peptides 134–157 and 183–207 formed two α-helices. Y185 and Y189, two residues of the catalytic triad, were located in the dimer interface, and the third one of the triad, S172, was nearby. This suggests that the formation of the HSD17B13 dimer is important to the proper formation of the catalytic center of the protein. The C-terminal helix-turn-helix motif contributed to the rest of the dimer interactions.

### Co-factor binding site and a blocked substrate tunnel in apo HSD17B13

The cofactor of HSD17B13, NAD⁺, was observed in all the crystals and was critical for crystallization. The nicotinamide nucleotide of NAD⁺ interacts with the putative catalytic triad of HSD17B13 (S172, Y185, and K189)[23] and a long loop in HSD17B13, loop P218-T239. The adenine nucleotide of NAD⁺ forms hydrogen bonds with side chain of D67 and D93, and the main chain of C94. Residues D67, D93-C94 are near or in peptide 71-106, which interestingly is deleted in an alternative splicing

isoform variant of HSD17B13, isoB. Given these missing residues, isoB variant is unlikely to have a properly formed cofactor binding site and an intact dimer interface. The isoB variant was localized in the ER but not at LD, probably due to protein misfolding and aggregation[11].

The interactions of NAD⁺ with HSD17B13 provide a structural explanation for the cofactor specificity of HSD17B13 (Fig. 3a). The side chain of a negatively charged residue, D67, interacts with the 2′ and 3′ hydroxyl of the adenine mononucleotide of the cofactor. The negative charge of D67 would be repulsive with the phosphate group if a NADP cofactor bound there instead. As a comparison, HSD17B1 is an NADP-specific enzyme, and it has been crystallized with NADP present enabling a comparison between cofactor interactions. In HSD17B1, L36 is present in place of D67 and faces away from the hydroxyls[38]. Next to L36 is R37, and the positive side chain of R37 forms ion pair interactions with the 2′-phosphate of NADP bound in HSD17B1. The equivalent of R37 in HSD17B13 is I68, a hydrophobic side chain that further disfavors NADP binding. The 2′-phosphate and 3′-hydroxyl groups of NADP in HSD17B1 were also recognized by S11, equivalent of which is G45 in HSD17B13.

A prominent feature of the apo HSD17B13 structure is that the putative substrate-binding site is completely blocked by the C-terminal peptide beyond the helix-turn-helix motif (Fig. 3b). This peptide, N286-K300 (C terminus), interacted extensively with the long loop P218-T239. Loop P218-T239 is equivalent to the substrate-binding loops of other HSD family proteins, and they were mostly disordered in the apo structures of those HSDs[39]. The C-terminal peptide of HSD17B13 additionally interacted with other parts of the protein. These interactions included hydrophobic contacts and hydrogen bonds involving main chain or side chain atoms. When a steroid molecule was docked in the active sites of the apo dog and human HSD17B13 by homology modeling (Fig. 3b), the molecule would clash with the C-terminal peptide and loop P218-239. In fact, repeated soaking with substrate

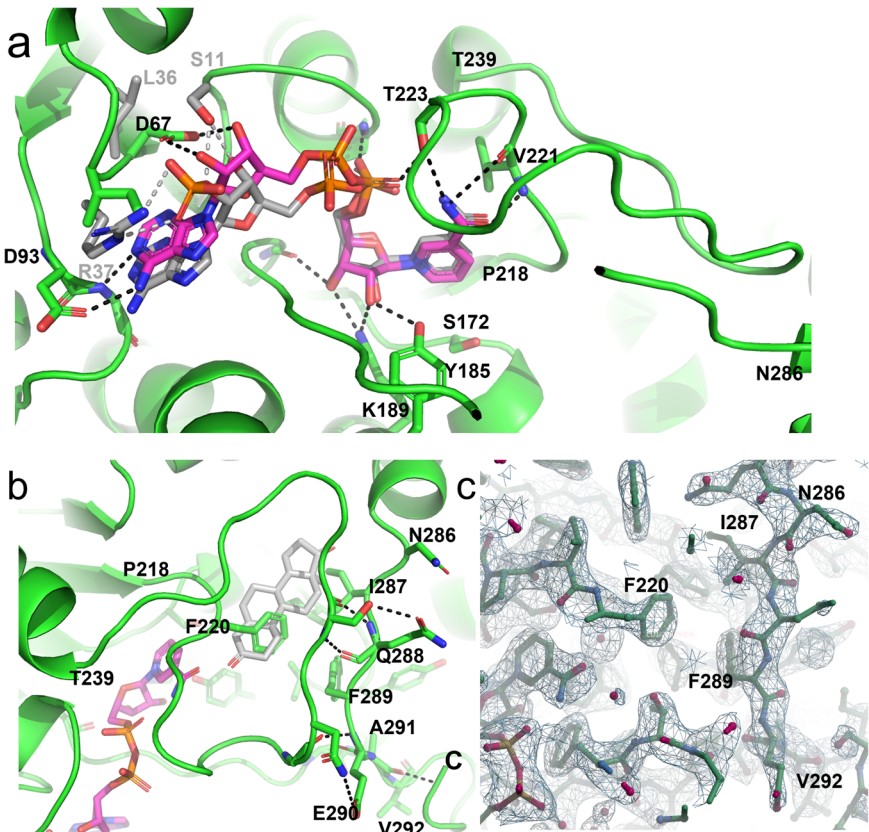

**Fig. 3 | Co-factor binding site and a blocked putative substrate site in apo HSD17B13 structure. a** The cofactor binding site of HSD17B13 selects for NAD$^+$(NADH). HSD17B13 residues making hydrogen bonds (dashes) with bound NAD$^+$ (stick model with carbon atoms colored magenta) were shown in sticks (green C atoms). As a comparison, the key residues that select for NADP in HSD17B1 (PDB ID 1QYV) were shown with gray C atoms. The putative catalytic triad of HSD17B13 S172, Y185, and K189 are shown and labeled. The substrate-binding loop, P218-T239, was labeled. **b** The C-terminal peptide of HSD17B13 occupied the putative substrate-binding site. Residues in the C-terminal peptide interacting with the P218-T239 loop were shown in sticks, and hydrogen bonds were shown in dashes. The putative substrate-binding site was indicated with an androstenedione molecule (spheres with C colored gray) docked from HSD17B1 complex structure (PDB ID 1QYX). **c** zoomed-in view of the electron densities (2Fo-Fc contoured at 1.2σ) surrounding the C-terminal peptide blocking the substrate site.

β-estradiol failed to result in ligand binding. Both sequence- and conformation-wise, the substrate-binding loops and the C-terminal peptides are the most variable in HSD proteins (Supplementary Figure 2, Supplementary Figure 4). As comparison, in the apo structure of HSD11B1, there was an empty cavity at the substrate site for access[35]. Substrate binding in HSD17B1 only caused minor changes at its substrate site[34]. The flexibility of the substrate-binding loops of other HSDs allowed substrates to readily access their active sites[39]. In many instances, the substrates could be soaked into crystals of those proteins.

The blockage of the substrate-binding site by the C-terminal peptide has two implications for the function of HSD17B13. First, filling the pocket with the C-terminal peptide stabilizes HSD17B13. As the isoD (P274del) variant lacks the C-terminal peptide as well as having an incompletely formed membrane interacting amphipathic helix, both factors likely contribute to isoD instability. Second, access to the substrate site needs to compete with the C-terminal peptide, which may reduce the catalytic efficiency of HSD17B13 with any substrate. Indeed, after screening recombinant HSD17B13 against 265 unique putative substrates including bioactive lipids and steroids, very few substrates were identified as has been previously reported[10]. When compared to HSD17B2 using β-estradiol, one of the most active substrates identified in the screen, the enzymatic activity of HSD17B13 was much lower than that of HSD17B2 (Supplementary Figure 5). In human HSD17B2 is widely expressed in many tissues, including liver[40]. Human HSD17B13 was also reported to have retinol dehydrogenase activity[11],

but multiple enzymes also efficiently oxidize retinol in liver[41,42]. Leukotriene B4, another one of the most active substrates identified, is also reported to be a substrate of other dehydrogenases[43]. The low efficiency of HSD17b13 towards these substrates when other dehydrogenases are present in liver, and are more efficient at catalyzing these reactions, begs the question of the biological relevance of HSD17B13 enzymatic activities of the substrates noted above.

**Complex structures of compound 1 with dog HSD17B13**

The overall structure of the compound **1** complex with dog mutant HSD17B13 did not have large changes in the catalytic cores of the HSD17B13 dimer compared to the apo HSD17B13 structure (0.31 Å for 437 Cα atoms of the dimer) (Fig. 4a). Compound **1** bound to both subunits of the dog mutant HSD17B13, displacing the C-terminal peptides from the substrate sites of both subunits. Missing the anchoring to the substrate sites from the C-terminal peptides, the adjacent P274-N286 helices in the helix-turn-helix motifs became disordered. Compound **1** binding also rearranged the P218-T239 loops, which interacted with the C-terminal peptides in the apo structure. Within the loop P229-L237 peptides became disordered, and the side chain of F220 adopted a different rotamer (Fig. 4b). The N-terminal helices packed against part of the helix-turn-helix motifs in the apo structure, and they became more flexible in the complex structure. Interestingly in the compound **1**-dog WT HSD17B13 crystal, **1** only bound one subunit of the HSD17B13 dimer, and the unoccupied subunit adopted the apo conformation. Because dog WT HSD binds **1** weakly (shown

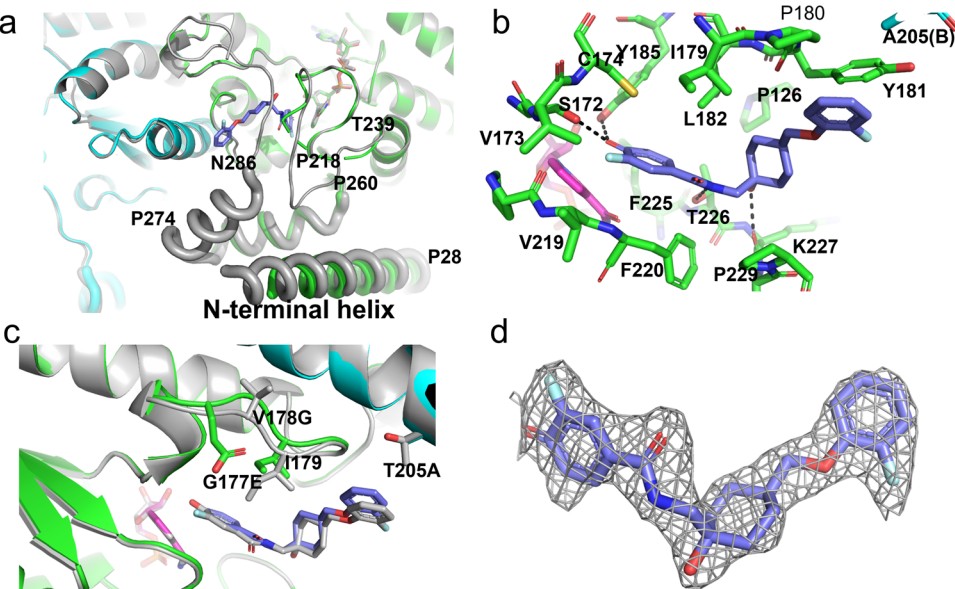

**Fig. 4 | Structure of dog HSD17B13/compound 1 complex. a** conformational changes of HSD17B13 caused by compound **1** (light blue C atoms, stick model) binding. The complex structure was superimposed with the apo dog HSD17B13 structure (gray ribbons). Key residues are labeled. **b** Detailed interactions of compound **1** with HSD17B13. Residues that contact compound **1** are labeled. **c** Local conformational changes at dog HSD17B13 brought by mutations into human amino acids (G177E/V178G/T205A). **d** To demonstrate the modeling quality, compound **1** was shown embedded in the initial 2Fo-Fc electron density map calculated before ligand was modeled.

below), subtle differences in the crystals could affect ligand binding, even though it crystallized in the same crystal form with the same crystal contacts as the dog mutant HSD, which binds **1** strongly.

At the ligand binding site compound **1** interacted with the catalytic center of the protein. The hydroxyl of the phenol made hydrogen bonds with the active residues S172 and Y185 (Fig. 4b). It contacted the positively charged nicotinamide ring of the bound cofactor, NAD$^+$, with the closest distance at 3.1 Å to the C4N atom. The fluorophenol of **1** bound to HSD17B13 was likely deprotonated, with the oxygen atom carrying negative charge. The pKa of the phenol could be lowered by fluorine and amide substituents. The amide and the hydroxyl of the cyclohexyl of **1** donated hydrogen bonds to the side chain of T226 and the main chain of K227, respectively. The non-polar atoms of **1** made extensive hydrophobic contacts with 12 residues of HSD17B13, including a π-stacking interaction between the terminal F-phenyl group and the side chain of Y181. All of the residues interacting with 1 were from one subunit, except A205 (human sequence), which was from the second subunit of the HSD17B13 dimer. The terminal fluorophenyl group of **1** extended toward a solvent-facing opening left by the displaced C-terminal peptide (Fig. 4a).

We compared the residues contacting **1** directly between the dog and human HSD17B13 sequences. They are all identical between dog and human HSD17B13, except 205, which is a Thr in dog and an Ala in human. However, there are several residues adjacent to **1** interacting residue that are different between human and dog HSD17B13, among them are 117/118, which are Gly/Glu in human and Val/Gly in dog. In the displaced C-terminal peptide 293 is a Val in human and an Ile in dog. The dog HSD17B13 mutant (V177G/G178E/T205A/I293V) had these residues changed from dog residues to human ones. These mutations did not change the overall structure of dog HSD17B13. The RMSD was 0.25 Å when 445 Cα atoms were superimposed between compound **1** dog WT and dog mutant HSD17B13 complex structures. However, local changes brought by these mutations were obvious (Fig. 4c). T205 A changed a larger polar residue in dog HSD17B13 to a smaller hydrophobic one in human protein, and the terminal F-phenyl group that contacted this residue adjusted position accordingly. The side chain of E177 (human) pointed toward the inhibitor binding site (but did not

have polar interactions with the inhibitor or the rest of the protein), and V178 (dog) pointed outward and pushed against the second subunit of the HSD17B13 dimer. The combined effects of G177E and V178G mutations made an outward shift of the 177-180 peptide away from the ligand, leaving more space at the active site of the mutated dog HSD17B13 for inhibitors and substrates. I293V was in the C-terminal peptide and disordered in the complex. As a result, **1** had full occupancies in both HSD17B13 subunits in the crystal of the mutant dog protein (Fig. 4d).

Of note, a phenol-containing compound, BI-3231, was recently reported to be a potent and selective human HSD17B13 inhibitor[44]. We docked BI-3231 into the active site of HSD17B13/**1** complex structure (Supplementary Figure 6). In the docked model, BI-3231 occupies a similar position as **1** in HSD17B13, and the di-fluorophenol of BI-3231 makes similar interactions with the active site residues and the bound cofactor NAD$^+$ as the fluorophenol of **1**.

## Complex structure of compound 2 with human mutant HSD17B13

The crystal of the compound **2**-human HSD17B13 contained four HSD17B13 dimer complexes in the asymmetric unit, with **2** occupying all the protein subunits. In some subunits, long rod-shaped electron densities were observed together with those of **2** at the binding sites (Fig. 5a). We built detergent C12E8 molecules into these densities, but these densities could be from the alkyl chain of the solubilized endogenous phospholipids as well. Additional detergent molecules were also seen around N-terminal helices of some HSD1713 subunits.

The overall structures of the catalytic cores of human HSD17B13 dimers upon **2** binding were essentially the same as those of the apo and the 1 bound dog HSD1713 (RMSD values ranged from 0.22-0.46 for 413-428 Cα atoms). Like **1**, **2** binding also displaced the C-terminal peptides in HSD17B13. In detergent co-existing binding sites, these peptides, as extensions of the P274-N286 helices, shifted positions relative to those of the apo dog HSD17B13 to interact with bound C12E8 molecules. In binding sites absence of detergents, the C-terminal peptides were disordered. Loops P218-T239 were rearranged relative to the apo structure and within them peptides P229-L233 were

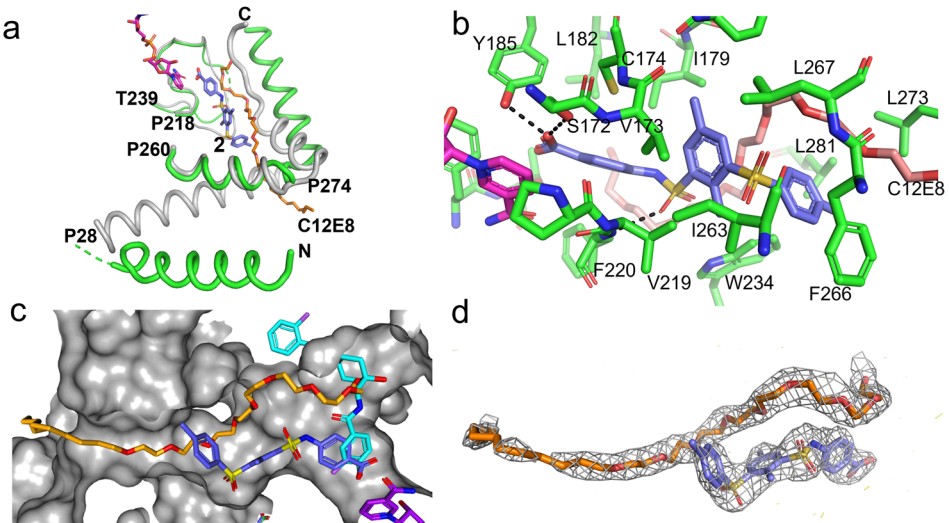

**Fig. 5 | Structure of human mutant HSD17B13/compound 2 complex.**
**a** Conformational changes of HSD17B13 caused by compound **2** (light blue C atoms, stick model) binding. Lipids/detergent molecules are also observed in the ligand binding site (orange carbon, stick model). For clarity, only the ligand binding site and the membrane anchoring domain from one subunit are shown. The complex structure was superimposed with the apo dog HSD17B13 structure (gray ribbons). Key residues are labeled. **b** Detailed interactions of compound **2** with HSD17B13. Residues that contacted compound **2** are labeled. Hydrogen bonds are in dashes. **c** Comparison of binding modes of compound **2** and compound 1 (cyan C atoms) at the ligand binding site of human HSD17B13/compound **2** complex (gray surface). The area of HSD17B13 with which the F-phenyl group of compound 1 clashes is from the extended helix, P274-A291. **d** To show the model quality, compound **2** was shown embedded in the initial 2Fo-Fc electron density map calculated before the ligand was included.

disordered. The N-terminal membrane anchoring helices of four HSD17B13 dimers also adopted variable conformations that are all different from those of the apo structure (Supplementary Figure 7). The variability of the membrane domains of human HSD17B13 dimer arises from the heterogeneity of the bound lipid/detergent molecules with which they interact. Most of these helices were still parallel to the lipid monolayer surface when modeled on LD surface, however, some of these helices tilted and reached deeper (~26 Å) into the interior of the membrane, likely deep enough into the neutral lipids core of the LDs. Nevertheless, none of these helices could cross an ER membrane bilayer, the typical thickness of which is $37.5 \pm 0.4$ Å[45].

Like 1, the negatively charged carboxylate of **2** formed hydrogen bonds with S172 and Y185 and contacted the positively charged nicotinamide ring of NAD[+] at closest distances of 3.1 Å to the C4N and C5N atoms (Fig. 5b). The conserved interaction with NAD[+], termed charge-transfer interaction, was well studied in NAD[+]-phenol model system[46]. One of the sulfonamide oxygen atoms formed a hydrogen bond with the main chain of F220. The terminal tosyl group extended toward the lipid membrane, interacting with the helix-turn-helix motif. The non-polar atoms of **2** made extensive contacts with hydrophobic side chains of 12 residues of HSD17B13, all from one subunit, as well as with the bound C12E8 molecules. There are distinct subsets of residues that contact only either **2** or **1**. The C12E8 molecules themselves extended beyond the helix-turn-helix motif, likely entered the interior of the lipid membrane (if on LDs) or detergent micelles (in solubilized states).

There are also differences in the binding poses of **1** and **2**. The carboxylate oxygen atom of 2 which formed hydrogen bonds with S172 and Y185 of HSD17B13 overlapped with the phenol oxygen atom of **1**, but the benzene rings shifted positions relative to each other, leading the rest of the molecules to interact with different parts of the binding site (Fig. 5c). While the terminal fluorophenyl of **1** pointed toward an opening that faces solvent, this opening was sealed by the extended P274-A291 helix in the **2** complex with detergent. The terminal tosyl group of **2** pointed toward an opening facing the membrane, through which the C12E8 molecule also exited. Nevertheless, like 1 in the dog mutant protein, compound **2** is well-ordered at the active site of human HSD17B13 (Fig. 5d).

## Structure-activity relationship studies

To better understand structure-activity relationships of the ligand binding site of HSD17B13, we compared the specific enzymatic activities of purified dog, human, and mutated HSD17B13 against β-estradiol (Fig. 6a). At all titrated enzyme concentrations human wild-type HSD17B13 was significantly more active than the dog wild-type protein. Mutating residues 177 and 178 in dog protein to match human protein (G177E, V178G) significantly enhanced enzymatic activity. Further mutating residues 205 and 293 to match the human sequence (T205A, I293V), matching what was used for crystallization, lead to an additional activity increase.

We next tested the cofactor selectivity in a biochemical assay using NAD[+] and NADP[+] as cofactors, and using β-estradiol and LTB4 as substrates, respectively. While NAD[+] could promote the oxidization of both substrates, NADP[+] did not achieve significant activity compared to the background signal cofactor turnover of inhibited reactions (Fig. 6b). Given that cytosolic NAD[+] concentration far exceeds those of NADH and NADP in liver[47,48], it is expected that HSD17B13 is mainly an oxidative enzyme using NAD[+] as cofactor.

Because of the low enzymatic activities of dog wild-type HSD17B13 (Fig. 6a) and HSD17B11 with β-estradiol, it was difficult to measure compound inhibition of these proteins in a biochemical assay and we instead utilized direct binding methods to compare compounds interaction with different proteins. To understand ligand binding, we tested several approaches such as surface plasmon resonance and isothermal calorimetry. Like many membrane/detergent-associated proteins, however, these methods were not amenable for HSD17B13. We next turned to thermal shift assays (TSA) for comparing ligand binding affinities[49]. TSA measures the changes in the thermal denaturation temperature of a protein under different conditions such as the presence compound, changes in buffer conditions, or sequence mutations. There are various readouts including intrinsic fluorescence or light scattering signals. We chose the static light scattering methods to detect the temperatures at which our proteins start to aggregate (Tagg), which are more suitable for membrane proteins that require detergent micelles for stability[50]. Given that in theory[51] and in practice Tagg continues to increase with concentration of ligands far

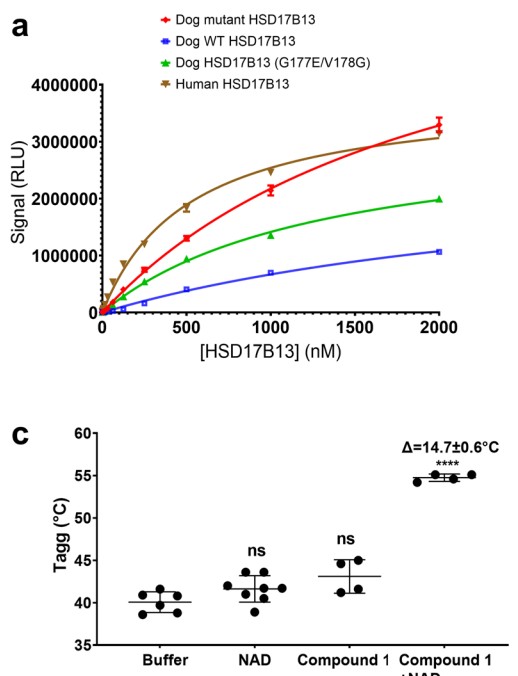

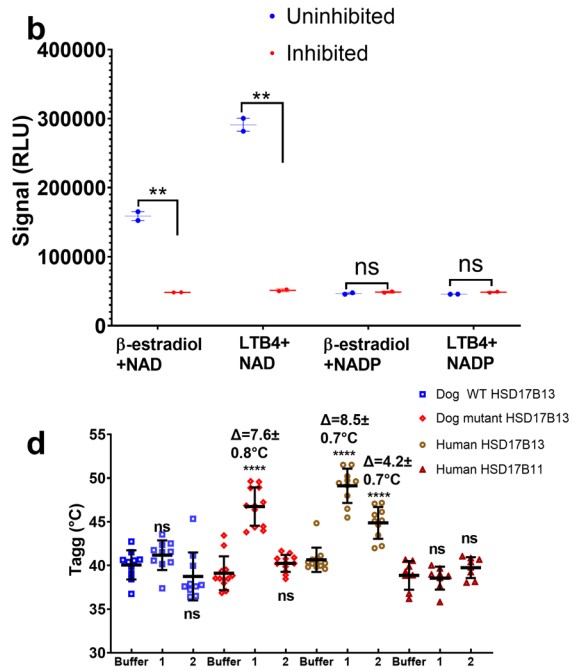

**Fig. 6 | Structure-activity relationship of HSD17B13. a** Representative enzymatic activity titration curves of human (brown triangles), dog WT (blue squares), dog quadruple mutant (G177E/V178G/T205A/I293V, red diamonds), and dog double mutant (G177E/V178G, green triangles) HSD17B13, using 12 μM β-estradiol as substrate and 500 μM NAD$^+$ as cofactor. The signals in relative luminescent units (RLU) were plotted against titrated enzyme concentrations, and non-linear regression fittings were also shown. The means of replicated measurements ± associated standard errors (SEM) were plotted ($n = 2$ independent experiments). Regression fitting and statistics in this figure were calculated using the program GraphPad[67]. **b** To test cofactor selectivity, NAD$^+$, and NADP were used as cofactors in the human HSD17B13 enzyme assay, using β-estradiol and LTB4 as substrates. The protein concentrations used were 20 nM. For inhibited reactions, 30 μM **1** was added to the reaction mixtures. The statistics were calculated between uninhibited (blue circles) and inhibited reactions (red circles) of the same cofactor and substrate ($n = 2$ independent experiments). Data were presented as mean values ± SEM. The unpaired, two-sided $t$ test $p$ values are 0.003 and 0.001, between uninhibited and inhibited reactions, for β-estradiol and LTB4 as substrates respectively, with NAD$^+$ as cofactor. The $p$ values are 0.29 and 0.05, respectively with NADP as cofactor. **c** Thermal shift assay demonstrating NAD$^+$-dependent binding of compound **1** to

human HSD17B13. The Tagg values in this figure were temperatures at which protein started to aggregate, recorded using static light scattering at 266 nm wavelength. The ΔTagg and statistics were calculated between 500 μM NAD$^+$ or compound 1 or compound **1** + 500 μM NAD$^+$ added samples and samples that contained only buffer plus DMSO ($n = 8$, 4, and 6 independent experiments). Data were presented as mean values ± SEM. The p values are 0.058, 0.045, and <0.00001, respectively. **d** Compound **1** and **2** binding to dog WT (blue squares), dog mutant (red diamonds), human HSD17B13 (brown circles), and human HSD17B11 (purple triangles) in the presence of 500 μM NAD$^+$. Data were presented as mean values ± SEM ($n = 10$, 10, 10, 12, 12, 12, 12, 10, 10, 8, 8 and 8 independent measurements, for buffer, 1 and 2, respectively, with dog WT, dog mutant, human HSD17B13 and human HSD17B11, respectively). ΔTagg and statistics were calculated between protein samples in the compound buffer and the same protein in the control buffer containing only DMSO. The $p$ values are 0.16 and 0.21; <0.0001 and 0.084; <0.001 and <0.001; and 0.7 and 0.23, for dog WT, dog mutant, human HSD17B13, and human HSD17B11, respectively. Unpaired two-sided $t$ test was used in statistics analysis. Statistics symbols: ****$p < 0.0001$; ***$p < 0.0005$; **$p < 0.005$; *$p < 0.05$; ns, $p > 0.05$.

above their IC$_{50}$s, we used high cofactor and inhibitor concentrations in the TSA experiments to maximize the ligand binding signal.

Because we observed direct interactions between **1** and **2** with bound NAD in crystals, for the TSA experiments we first tested if binding of compound **1**, the more potent of the two, to HSD17B13 is NAD$^+$ dependent. While compound **1** resulted in a significant increase in thermal stability for human HSD17B13 in the presence of NAD$^+$, there was no significant effect in the absence of cofactor (Fig. 6c). This was consistent for three additional close analogs of **1** (Supplementary Figure 8). There was also a strong linear relationship between ΔTagg of **1** and its analogs, and the logarithm of their biochemical IC$_{50}$ values when the compound concentrations were kept the same and the same batch of protein was used (Supplementary Figure 6), consistent with thermodynamic analysis that strong binders have larger stabilization effects on proteins[52]. This calibration method of using multiple compounds is a better way of correlating the ΔTagg and IC$_{50}$ values than dose response of a single ligand, because the K$_D$ derived from TSA may vary with ligand concentration[51].

Finally, we measured the binding of compound **1** and **2** to dog WT, dog mutant HSD17B13, human HSD17B13, and HSD17B11 (Fig. 6d). Neither **1** nor **2** stabilized dog WT HSD17B13 significantly. Inability to

detect stabilization of dog HSD17B13 by 1 in TSA does not mean **1** does not bind dog HSD17B13. In our experience TSA is insensitive to detect weak binders (>10 μM K$_d$) for some proteins. This might also explain why compound **1** only occupied one subunit of the HSD17B13 dimer in the dog WT HSD17B13 complex crystals despite the use of 1 mM of compound **1** in co-crystallization, and compound **2** failed repeatedly in soaking into these crystals. Compounds **1** and **2** bound human HSD17B13 strongly. Dog mutant HSD17B13 bound compound **1** strongly, and only slightly less than the human HSD17B13 (dog mutant HSD1713/compound **1** Tagg: 46.8 ± 2.2 °C; human HSD17B13/compound **1** Tagg: 49.1 ± 2.0 °C; unpaired $t$ test $p = 0.016$), indicating that the mutations introduced in dog HSD17B13 based on compound **1** interactions increased its binding affinity. Dog mutant HSD17B13 did not bind compound **2** significantly, and significantly less than the human HSD17B13 (dog mutant HSD17B13/**2** Tagg: 40.2 ± 0.97 °C, human HSD17B13/**2** Tagg: 44.9 ± 1.8 °C, $p < 0.0001$). This mutant did not completely address residue differences at the compound **2** site of human and dog HSD17B13. For example, compound **2** contacts I263 and F266 of human HSD17B13, which are both Tyr in dog HSD17B13. Finally, when tested with human HSD17B11, neither **1** nor **2** showed an effect on protein stability. Indeed, there are several residues that

contacted both compounds **1** and **2** in HSD17B13 that are different in HSD17B11, for example, C174, I179, and V219 in HSD17B13, which are Ala, Val, and Asn, respectively.

## Discussion

Overall, our HSD17B13 structures suggest a mechanism for LD association where the protein embeds on top of the phospholipid membranes and the membrane anchoring helices do not cross lipid bilayers. The stretch of hydrophobic residues at the N-terminus and the key P28 are all essential for HSD17B13's association with LD. Residues 1–28 are also necessary and sufficient for targeting HSD17B11 to LD[28]. Residues 22-28 of HSD17B13 share weak sequence conservation with the PAT domain[11], a N-terminal domain of many well-known LD-associated proteins, such as Perilipin, ADRP, and TIP47[24,25]. Inspection of PAT domain structures predicted by AlphaFold 2[31] showed conserved prolines at positions of turns in a layered amphipathic helix-turn-helix-turn-helix-turn-helix motif. The PAT domain of these proteins should structurally function similarly to the membrane-associating domain of HSD17B13, except in HSD17B13 the peptides in the membrane-associating domain are discontinuous in primary sequence (1–28 and 260-286). To the best of our knowledge, the only previously disclosed structure of a LD anchoring peptide was the NMR structure of the membrane anchoring motif of CGI-58 (PDB ID: 5A4H), which was largely unstructured[53]. Although the N-terminal peptide 1-28 is the main membrane anchoring element of HSD17B13, the amphipathic helix-turn-helix 260-286 is also important for the stability of the protein on the surface of lipid droplet. Disruption of the helix-turn-helix likely results in protein misfolding/instability and low protein level in cells, as is in the case of the P260, P274del, and exon 6 skipping variants of HSD17B13.

Observations of lipid/detergent molecules in human HSD17B13/**2** complex interacting with the N-terminal helices and the helix-turn-helix motifs directly demonstrated that these structural elements are indeed responsible for membrane anchoring. They also explain the effects of detergents on the stability of HSD17B13. These molecules were visible in human HSD17B13 crystals likely because they were ordered by their interactions with both the protein and the bound **2**. On the other hand, they were disordered in the dog HSD17B13 crystals, although they were in the protein samples used for crystallization. We could not observe **2** with dog HSD17B13 crystals because of its weak binding affinity. The apo HSD17B13 has essentially the same structure in different crystals, thus it may closely resemble the protein on the surface of the lipid-droplet in the absence of inhibitors. The ligand binding site and the membrane anchoring domain may transiently sample other conformations due to their dynamic nature and fluidity of the membrane, allowing the lipid molecules in the lipid-droplet transiently access the active site of HSD17B13.

Our structure-activity studies indicated that the differences in amino acids at the ligand binding sites (dog vs human HSD17B13, human HSD17B13 vs HSD17B11) affect the activities of both the inhibitors and the substrate tested. Interestingly, in mice HSD17B13 deficiency did not protect against liver injury[54], instead, it triggered hepatic steatosis and inflammation[55]. Among residues that were shown in our studies to be important for substrate and **1** activities, three residues are different between human HSD17B13 (E177, G178, and A205) and mouse HSD17B13 (G177, V178, and T205). For **2**, in addition to residues 177 and 178, residues 266 and 281 are different between human and mouse HSD17B13 (Phe and Leu in human vs Ser and Ile in mouse, respectively). These amino acid differences in the substrate-binding site of HSD17B13 between human and mouse may result in differences in substrate specificity and explain the difference in pathogenicity of this protein in human and rodents[54,55]. In fact, species differences in substrate selectivity and inhibitor binding have been observed for other HSD family members[56–58].

The observation of a completely blocked substrate site in the apo HSD17B13 structure may explain the challenges we and other groups faced in identifying active substrates[10]. While potent inhibitors could displace the C-terminal peptides, endogenous bioactive molecules may transiently but much less efficiently access the active site of HSD17B13 and get turned over. The observed low basal enzymatic activity of the wild-type human HSD17B13 together with data that reported protective variants have both diminished enzymatic activities and lower protein expression levels, raises the possibility that the pathogenicity of human HSD17B13 comes from the scaffolding function, rather than from the enzymatic activity alone, of the protein on the LD surface. Recently HSD17B13 was reported to bind adipose triglyceride lipase (ATGL), the rate-limiting lipolytic enzyme in triacylglycerol hydrolysis, involving a conserved serine residue, S33[59]. Phosphorylation (not observed in our samples) of S33 of HSD17B13 attenuates, while S33A mutation exacerbates NAFLD in mice[59]. S33 locates in the linker peptide that connects the N-terminal membrane anchoring helix and the catalytic domain. In our model it is above the membrane surface and exposed to aqueous environment so it can interact with ATGL, a soluble protein. A S33A mutation will make it hydrophobic, and phosphorylation adds a large, charged group to it. If the scaffolding function of HSD17B13 is important, then approaches of lowering HSD17B13 protein levels are needed, such as antisense or short interfering RNA[60,61].

Altogether, our crystal structures provide important information for developing novel HSD17B13 binders. The observation of compound **1** and **2** occupying different parts of the HSD17B13 ligand binding site in our crystal structures further indicate that diverse chemical motifs can bind to HSD17B13, and multiple series of compounds could be developed as HSD17B13 inhibitors.

## Methods
### Constructs

For protein and membrane purification, the following constructs were cloned into pFastBac unless otherwise indicated. For human HSD17B13 and HSD17B11, from N- to C-terminus: 10× His, a biotin acceptor peptide (GLNDIFEAQKIEWHE), a TEV protease site, and either human HSD17B13 (residues 2-300, NP_835236.2) or human HSD17B11 (residues 2-300, NP_057329.3). For dog HSD17B13 constructs, from N- to C-terminus: HSD17B13 (AA1-300, XP_038300006.1), a Gly-Ser-Gly linker, and a 11× His. For human HSD17B13 a similar version was made in pcDNA3.1 for mammalian overexpression. For substrate selectivity studies, the following combination of point mutations was made to dog HSD17B13: G177E/V178G or G177E/V178G/T205A/I293V. For human HSD17B13 crystallization, the following construct was made from N- to C- terminus: Human HSD17B13 (residues 2-300, NP_835236.2) with or without Q60K/I62R/R71H/E161K point mutations, a Gly-Ser-Gly linker, and 11× His. Proteins were expressed in *Spodoptera frugiperda* (Sf9, ATCC) using the Bac-to-Bac Baculovirus Expression System (Invitrogen) with infection at a cell density of $2 \times 10^6$ cells ml$^{-1}$ for 72 hrs. For human HSD17B2, the construct was cloned instead into pcDNA3.1 and contained from N- to C-terminus: 6xHis, a TEV protease site, and human HSD17B2 (residues 2-387, NP_002144.1). The protein was expressed in Expi293 Cells (Life Technologies) grown in Expi293 Expression media to densities of $2.5 \times 10^6$ cells ml$^{-1}$. Cells were transiently transfected for 72 hrs with 1 μg plasmid per mL cell ratio in Opti-MEM reduced serum media (Life Technologies) using ExpiFectamine Transfection Kit (Life Technologies). Control and human HSD17B13 Expi293 membranes were similarly made using cells transfected with empty pcDNA3.1 plasmid pcDNA3.1 plasmid encoding HSD17B13.

### Protein and membrane purification for biochemical evaluation

For HSD17B2 and control membranes, transfected Expi293 cells were resuspended in 20 mM sodium phosphate pH 7.2, 20 mM NaCl, 250 mM sucrose, 10% glycerol, 1 mM TCEP, 0.5 mM NAD$^+$, cOmplete

EDTA-free protease inhibitors (Roche), and Benzonase (MilliporeSigma). Cells were lysed using a microfluidizer, spun at 500 × g for 10 min to remove unlysed cells, and membranes were isolated by ultracentrifugation.

For purified human HSD17B13 and HSD17B11 protein used for biochemical evaluation, Sf9 cells were resuspended in 50 mM sodium phosphate pH 7, 500 mM NaCl, 10% glycerol, 1 mM TCEP, cOmplete EDTA-free protease inhibitors (Roche), and Benzonase (MilliporeSigma). Cells were lysed using a microfluidizer and pelleted by ultracentrifugation. Membranes were solubilized for 4-6hrs at 4°C in 50 mM NaPhos pH 7.2, 300 mM NaCl, 50 mM Imidazole, 10% glycerol, 1 mM TCEP, 0.5 mM NAD$^+$, and 0.5% DDM (Anatrace). The DDM-solubilized fraction was clarified by ultracentrifugation and incubated overnight at 4°C with NiNTA agarose (Qiagen). Resin was washed through a series of steps with 50 mM sodium phosphate pH 7, 150 mM NaCl, 10% glycerol, 1 mM TCEP, 50 μM NAD$^+$ and (1) 50 mM Imidazole, 0.5% DDM, (2) 50 mM Imidazole, 0.1% DDM, (3) 50 mM Imidazole, 0.05% DDM, and (4) 250 mM Imidazole, 0.02% DDM, and eluted with (6) 150 mM NaCl, 700 mM Imidazole, 0.02% DDM. Finally, the protein was further purified by S200 10_300 gel filtration chromatography in 25 mM sodium phosphate pH 7, 150 mM NaCl, 10% glycerol, 1 mM TCEP, 50 μM NAD$^+$, and 0.02% DDM and concentrated in a 100 kDa MWCO centrifugal concentrator (Amicon).

For biochemical evaluation of dog HSD17B13 constructs, the purification was similar to above with a few changes. HEPES pH 7.4 buffer was used in place of sodium phosphate pH 7.0. Membranes were solubilized with 0.5% octaethylene glycol monododecyl ether (C12E8, Anatrace) instead of 0.5% DDM, and nickel resin was washed with decreasing [C12E8] to a final concentration of 0.005% C12E8. The final NiNTA fraction was run by S200 10_300 gel filtration chromatography in 50 M HEPES pH 7.4, 300 mM NaCl, 10% glycerol, 1 mM TCEP, 50 μM NAD$^+$, and 0.005% C12E8.

## Protein purification for crystallization

Like the above, SF9 cells were collected and resuspended in buffer containing 50 mM HEPES pH 7.4, 500 mM NaCl, 10% glycerol, 1 mM TCEP, EDTA-free cOmplete Protease Inhibitor cocktail (Roche), and Benzonase (MilliporeSigma). Cells were lysed using a microfluidizer and membrane was isolated by ultracentrifugation. The membrane pellet was resuspended in 50 mM HEPES pH 7.4, 300 mM NaCl, 50 mM Imidazole, 10% glycerol, 1 mM TCEP, 0.5 mM NAD$^+$ and solubilized by addition of 0.5% C12E8 (Anatrace) for 4 hours at 4 °C. The sample was clarified by ultracentrifugation, and the supernatant was batch bound to NiNTA agarose (Qiagen) overnight at 4 °C. The resin was washed in solubilization buffer containing 50 μM NAD$^+$ with stepwise decrease to 0.005% C12E8. The eluted protein was further purified on by SEC in 50 mM HEPES pH 7.4, 300 mM NaCl, 50 μM NAD$^+$, 10% glycerol, 1 mM TCEP, 0.005% C12E8. HSD17B13-containing fractions were pooled and concentrated in a 100 kDa MWCO centrifugal concentrator (Amicon).

## Inhibitors preparation and characterization

The preparation and characterization of compound **1** and its three analogs studied were reported before[29]. Compound **2** was repurified from internal collection and characterized using NMR and high-resolution mass spectrum (Supplementary Methods).

## Biochemical assays

HSD17B13 enzyme inhibition potency of test compounds was determined using a purified protein biochemical enzyme activity assay, using NAD(P)H-Glo luciferase readout (Promega). HSD17B13 enzyme uses the oxidized form of nicotinamide adenine dinucleotide (NAD$^+$) as a cofactor during metabolism of β-estradiol (substrate) to estradiol (product), while converting NAD$^+$ to the reduced form (NADH). The Promega NAD(P)H-Glo™ assay is a homogeneous bioluminescent assay that generates a light signal from biochemical reactions that contain NADH (or nicotinamide adenine dinucleotide phosphate, NADPH). In the presence of NADH (or NADPH), the enzyme Reductase reduces a proluciferin reductase substrate to form luciferin. Luciferin then is quantified using Ultra-Glo™ Recombinant Luciferase (rLuciferase), and the light signal produced is proportional to the amount of NAD(P)H in the sample. Substrate mix composed of 12 μM final assay concentration (FAC) β-estradiol (Sigma, E8875) and 500 μM FAC NAD$^+$ (Sigma, N8285) in assay buffer (25 mM Tris-HCl (Sigma T2444) and 0.02% Triton X-100, pH 7.6) was added (2 μL/well) to 384-well assay plates (Corning 3824) containing 80 nL of 50× FAC compound, serial diluted 1 in 3.162 in 100% DMSO for an 11 point concentration response curve (80 μM top concentration), with duplicate points at each concentration. The reaction was initiated by the addition of 2 μL/well purified HSD17B13 protein (30 nM FAC in assay buffer). Compound, substrate mix, and HSD17B13 protein were incubated in the dark at room temperature for 2 hours before the addition of 3 μL/well NAD(P)H-Glo detection reagent, prepared from luciferase detection reagent and reductase/reductase substrate, as per manufacturer's instructions (Promega, G9061). Detection reagent was incubated in the dark for 1 hour at room temperature before plates were read on the Envision plate reader (Perkin Elmer) using a luminescent protocol. Data expressed as relative luminescent units (RLUs) were then normalized to control wells using Activity Base (IDBS). Zero percent effect (ZPE) was defined as RLUs generated from uninhibited HSD17B13 protein (vehicle control). One hundred percent effect (HPE) was defined RLUs generated from wells containing 40 μM FAC of a Pfizer proprietary compound known to cause 100% inhibition of HSD17B13 protein. The concentration and % effect values for each compound were plotted by Activity Base using a four-parameter logistic dose-response equation, and the concentration required for 50% inhibition (IC$_{50}$) was determined.

As a technology counter screen, compounds were incubated along with 1 μM (final) NADH standard, without enzyme, to generate a signal, and then incubated with NAD(P)H-Glo detection reagent, with the purpose of identifying compounds that interfered with, and decreased, the NAD(P)H-Glo signal.

Dependency of co-factor (NAD$^+$ and NADP$^+$) was determined using a purified protein HSD17B13 biochemical enzyme activity assay, using NAD(P)H-Glo luciferase readout (Promega). Substrates β-estradiol (Sigma, E8875) and Leukotriene B4 (LTB4, Cayman), 15 μM FAC in assay buffer (25 mM Tris-HCl (Sigma T2444) and 0.02% Triton X-100, pH 7.6), and 500 μM FAC co-factor (NAD$^+$ or NADP, Sigma, N8285) in assay buffer were added (2.5 μL/well each) to 384-well assay plates (Corning 3824) containing 2.5 μL of either 30 μM FAC inhibitor in 1% DMSO/assay buffer or vehicle (1% DMSO/assay buffer), in triplicate. The reaction was initiated by the addition of 2.5 μL/well-purified HSD17B13 protein (20 nM FAC in assay buffer). Compound, substrate, co-factor, and HSD17B13 protein were incubated in the dark at room temperature for 2 hours before the addition of 10 μL/well NAD(P)H-Glo detection reagent, prepared from luciferase detection reagent and reductase/reductase substrate, as per manufacturer's instructions (Promega, G9061). Detection reagent was incubated in the dark for 1 hour at room temperature before plates were read on the Envision plate reader (Perkin Elmer) using a luminescent protocol. Data expressed as relative luminescent units (RLUs) were exported from the plate reader and plotted in GraphPad Prism.

The activities of membranes containing overexpressed HSD17B13 and HSD17B2 were measured in the same assay format. Co-factor NAD$^+$ (Sigma, N8285), prepared in assay buffer (25 mM Tris-HCl (Sigma T2444) and 0.02% Triton X-100, pH 7.6) at 2 mM final assay concentration (FAC), was added (2.5 μL/well) to 384-well assay plates (Corning 3824) along with either 2.5 μL of 50 μM FAC β-estradiol (Sigma, E8875) in assay buffer/2.5% DMSO, or vehicle (2.5% DMSO/assay buffer), in duplicate. The reaction was initiated by the addition of 5 μL/well membrane titration (HSD17B13, HSD17B2, or pcDNA3.1 blank)

in assay buffer. Substrate, cofactor, and membrane were incubated in the dark at room temperature for 1 hour before the addition of 10 μL/well NAD(P)H-Glo detection reagent, prepared from luciferase detection reagent and reductase/reductase substrate, as per manufacturer's instructions (Promega, G9061). Detection reagent was incubated in the dark for 1 hour at room temperature before plates were read on the Envision plate reader (Perkin Elmer) using a luminescent protocol. Data expressed as relative luminescent units (RLUs) were exported from the plate reader into excel, and background (non- β-estradiol wells) was subtracted from the wells containing β-estradiol, for a change in signal (delta, or Δ). Background subtracted data were plotted in GraphPad Prism, vs membrane concentration (μg/well) on the x-axis.

The activities of dog WT, dog mutant, and dog G177E/V178G HSD17B13 proteins were tested in the same assay format as the human HSD17B13. Co-factor and substrate mix containing 12 μM final assay concentration (FAC) NAD$^+$ (Sigma, N8285) and 12 μM FAC beta-estradiol (Sigma, E8875) in assay buffer (25 mM Tris-HCl (Sigma T2444) and 0.02% Triton X-100, pH 7.6) (2.5% DMSO final) was added (5 μL/well) to 384-well assay plates (Corning 3824). The reaction was initiated by the addition of 5 μL/well purified protein titration (wild-type dog, dog mutant and additional G177E/V178V dog mutant, and wild-type human HSD17B13) in assay buffer. Substrate, co-factor and membrane was incubated in the dark at room temperature for 2 hours before the addition of 10 μL/well NAD(P)H-Glo detection reagent, prepared from luciferase detection reagent and reductase/reductase substrate, as per manufacturer's instructions (Promega, G9061). Detection reagent was incubated in the dark for 1 hour at room temperature before plates were read on the Envision plate reader (Perkin Elmer) using a luminescent protocol. Data expressed as relative luminescent units (RLUs) were exported from the plate reader into excel, and plotted in GraphPad Prism, vs. enzyme concentration (nM).

## Cell assay

Cell inhibition potency of test compounds was determined using a whole-cell HEK overexpressing HSD17B13, utilizing a liquid chromatography/mass spectrometry (LCMS) readout.

In this assay HSD17B13 uses NAD$^+$ as a cofactor during metabolism of β-estradiol, converting NAD$^+$ to NADH and β-estradiol to estrone. Estrone production is quantified by LCMS using a Sciex API-5500-Electrospray system (AB Sciex LLC, MA, USA), and used as a measure of HSD17B13 enzyme activity.

HEK- cells stably expressing wild-type human HSD17B13 were plated at 10,000 cells/well in 50 μL growth media (DMEM containing 10% heat-inactivated FBS, 400 μg/ml geneticin, 1× L-Glutamine, and 1× non-essential amino acids, Invitrogen 11965-092, 16140-071, 10131-027, 25030-081, 11140-050), into poly-D-lysine-coated 384-well plates (Corning Biocoat, 354663), and incubated overnight (with lid) at 37 °C (95% O2: 5% CO2). Following overnight incubation, intermediate compound plates (Greiner, 781280) containing 160 nL of 375× FAC test compound which had been serial diluted 1 in 3.162 in 100% DMSO for an 11 point concentration response curve, with duplicate points at each concentration, were diluted 1 in 187.5 with 30 μL warmed assay media (DMEM, 1× L-Glutamine, and 1× non-essential amino acids) to give 2× FAC compound (80 μM top concentration) in 0.53% DMSO. Growth media was then removed from the cell plates and replaced with 10 μL of 2× FAC test compound and incubated for 1 hour (with lid) at 37 °C (95% O2: 5% CO2), before the addition of 25 μM FAC beta-estradiol in assay media/0.2% DMSO. The reaction was incubated for 2 hours (with lid) at 37 °C (95% O2: 5% CO2), after which 10 μL of reaction was transferred from assay plate to a new 384-well deep-well plate (Matrix 4325) and diluted 1 in 10 with 90 μL of stop reagent (50% methanol in water containing internal standard 17β-estradiol-2,3,4-$^{13}$C3). The amount of product

(estrone) was then quantified by LCMS using software provided (Analyst 1.7, Hotfix 3, MultiQuant 3.0.31721.0). Data expressed as product area ratio (PAR) were then normalized to control wells using Activity Base (IDBS). Zero percent effect (ZPE) was defined as PAR generated from uninhibited HSD17B13 (vehicle control). One hundred percent effect (HPE) was defined as PAR generated from wells containing 20 μM FAC of a Pfizer proprietary compound known to cause 100% inhibition of HSD17B13. The concentration and % effect values for each compound were plotted by Activity Base using a four-parameter logistic dose-response equation, and the concentration required for 50% inhibition (IC$_{50}$) was determined.

## Thermal shift assay

TSA studies were run on the UNit instrument (Unchained Labs, CA, USA). Experiments were run in either 25 mM sodium phosphate pH 7, 150 mM NaCl, 10% glycerol, 1 mM TCEP, 0.05% DDM, + different cofactors (NAD$^+$, NADH, NADP, NADPH; MilliporeSigma) or 25 mM HEPES pH 7.2, 300 mM NaCl, 0.5 mM TCEP, 0.05% DDM, 10% glycerol, ±500 μM NAD$^+$, and ±100 μM compound **1** (from 2 mM DMSO stock) or DMSO or 50 mM Hepes pH 7.5, 300 mM NaCl, 0.5 mM TCEP, 0.05% DDM, 500 μM NAD$^+$, ± 50 μM compounds (from 2 mM stock) or DMSO. Protein aggregation was measured by static light scattering at 266 and 473 nm with a 1 °C/sec ramp rate from 20-80 °C. The data was processed using the UNcle software (Unchained Labs, CA, USA), and temperatures at which the protein started aggregation were recorded as Tagg. Reported Tagg and analysis were based on measurements at 266 nm unless otherwise indicated. The protein concentration used was 10.4 μM.

## Protein crystallization

For the canine proteins, protein was incubated at 12 mg/ml with 1 mM NAD$^+$ and 1 mM compound **1** with the addition of 0.125% β-octyl-glucoside. Sitting drop vapor diffusion crystallization was set up by mixing 300 nl protein complex with 300 nl of reservoir solution containing 30% PEG3350, 0.2 M ammonium chloride. Crystals grew at room temperature over 2 weeks.

Human protein was crystallized with 1 mM NAD$^+$ and 1 mM compound **2** with the addition of 0.5% β-octyl-glucoside. Vapor diffusion was carried out in the same fashion as the dog HSD17B13 complex with a reservoir containing 25% PEG3350, 0.1 M Bis-Tris pH 5.5, 0.2 M Li$_2$SO$_4$. Crystals grew at room temperature over 2 weeks.

## Crystallographic data collection, structure determinations, and refinements

Crystal data sets were collected at beamline 17-ID at IMCA CAT, APS (Argonne National Laboratory, Chicago). The wavelength used was 1 Å for all crystals. Crystals were kept at 100 K during data collection using cryo stream. Data were processed using the program autoPROC utilizing ingratio n program XDS[62]. Data sets were scaled and merged using anisotropic resolution cutoff, and ellipsoidal data completeness were reported (Table 1).

The initial dog WT HSD17B13/compound 1 complex structure was solved with molecular replacement method using the published HSD17B11 crystal structure as model (PDB ID 1YB1), using the program Phaser[63]. Subsequent dog HSD17B13 apo/complex structures were solved using rigid body refinements. The compound 2/human HSD17B13 complex structure was solved with molecular replacement method using refine dog HSD17B13 structure as the starting model. Structure refinements were carried out using the program Buster[64] and manual model building using the program COOT[65]. Final refinement statistics are listed in Table 1. For the preparation of structural figures in this manuscript the graphic program PyMol was used[66].

**Reporting summary**

Further information on research design is available in the Nature Portfolio Reporting Summary linked to this article.

## Data availability

The atomic coordinates and structure factors have been deposited in the Protein Data Bank, Research Collaboratory for Structural Bioinformatics, Rutgers University, New Brunswick, NJ (http://www.rcsb.org/) under the accession codes 8G84, [https://doi.org/10.2210/pdb8G84/pdb], 8G89, 8G93 and 8G9V. Source data are provided with this paper.

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

## Acknowledgements

We thank Suman Shanker for technical assistance in protein purification, and Mary A. Piotrowski for help in cell assay.

## Author contributions

S.L. and R.S. designed and performed the experiments and wrote the manuscript. N.N., L.S., Y.W., and M.G. performed experiments and wrote the manuscript. J.D., L.Z., and D.E. performed experiments. M.C. wrote the manuscript.

## Competing interests

The authors declare no competing interests.
