## [Peer Review File · Nature Communications]

REVIEWER COMMENTS

Reviewer #1 (Remarks to the Author):

In this manuscript the authors elegantly show the crystal structure HSD17B13 (using dog, human, and hybrid constructs) in the presence of inhibitors and NAD⁺ cofactor. Key findings are the anchoring mechanism to lipid droplet membrane, details of the dimerization interface, and importantly, insights from the structure into the enzymatic activity, species differences and possible function of HSD17B13. The manuscript is very well written, the results and predictions are carefully presented and discussed, and the reported crystal structure is potentially a great resource for testing biological substrates and designing small molecule inhibitors of HSD17B13, which could have direct clinical implication. The findings are important and worthy of publication.

However, some concerns need to be addressed.

Major comments:

1. Identification of inhibitors: Although the inhibitors Compound 1 and 2 are not the main focus of the manuscript, the authors should provide information on the compound library, the methodological details of how the inhibitors were selected, as well as the direct results of the luciferase assays testing them. More importantly, since the inhibitor screening assay appears to be based on purified protein, it is critical to at least verify whether the selected inhibitors indeed inhibit HSD17B13 enzymatic activity in mammalian cells.
2. Both inhibitor compounds were selected from the inhibitor screening using purified human HSD17B13 (hHSD) protein, and compound 1 is the stronger binder to hHSD and reported to be a stronger inhibitor. In contrast, compound 1 does not bind dog HSD17B13 (dHSD) in the TSA assay and its inhibitory capacity for dHSD is not reported. It is therefore unclear why hHSD could not crystalize with compound 1 while dHSD could. Can the authors explain this apparent paradox? They should also report whether the compounds are indeed inhibitors of dHSD. Since the crystal structure of hHSD with compound 2 was a success, why not using compound 2 for the later thermal stability testing?
3. Please provide potential explanations for human HSD17B13 forming four dimers with compound 2. Is it WT hHSD or mutant hHSD (p. 7, line 2 and Table 1)? It is unclear from the preceding paragraph.
4. Stemming from comments 2 and 3 – hHSD was crystalized with the weaker inhibitor (compound 2) in 4-dimer configuration; dHSD was crystalized with compound 1 (which doesn't bind it well) in a single dimer with only one inhibitor. This raises a concern that the compounds themselves affected, and may have altered, the crystal structure and hence, that the findings are not truly reflective of the "true" membrane-bound structure in vivo. The authors should discuss extensively why they think compounds 1 and 2 are necessary for crystallization, and the possibility of this generating a major artefact.
5. Enzymatic activity of native HSD: Page 11 line 2-5, "When a steroid molecule was docked in the active site of the apo HSD17B13 by homology modeling (Figure 3B), the molecule would clash with the C-terminal peptide and loop P218-239. In fact, repeated soaking with substrate β -estradiol failed to result

in ligand binding.” Which species of HSD17B13 was used? How do the authors reconcile this finding of blocked substrate site with the fact that they, and others, were still able to demonstrate enzymatic activity against candidate substrates (even if these may not be the physiological substrate)? Does that support the authors’ interpretation of their findings to suggest a scaffolding, rather than enzymatic, role for HSD17B13?

6. The authors identify the c-terminal helix-turn-helix domain as a second component for anchoring HSD17B13 to the LD, together with the n-terminal helix. However, Ma et al (reference 11) previously showed that despite their instability, P260S, P274del and exon 6 skipping mutants (all expected to impact the 260-286 domain) do localize to the surface of lipid droplets in mammalian cells. Can the authors reconcile this with their structure predictions?

7. In Supplemental Figure 6, the ligand to protein ratio was 10:1 and the NAD to protein ratio was 50:1, which are far beyond physiological levels. Can the authors explain why they used an inhibitor concentration (100 μ M) which is markedly higher than the IC_{50} (IC_{50} ranging from 0.18 to 4.55 μ M), and why they used 500 μ M of the cofactor NAD?

8. In Supplemental Figure 6, the authors show a correlation between θ Tagg and IC_{50} of multiple inhibitors, all used with at the same concentration. Isn’t that effectively a surrogate for a dose response curve with a single inhibitor? We suggest the authors show a dose response curve instead.

9. Su et al recently (reference 57) demonstrated an impact of S33 phosphorylation on liver phenotype. This serine is located just beyond the n-terminal helix that is anchored in the LD monolayer. Even if not tested experimentally, can the authors model the effect of this phosphorylation on their predicted structure?

10. Thamm et al (J Med Chem, 2023) recently published a selective HSD17B13 inhibitor (BI-3231). Can the authors model its docking? If they show that BI-3231 is predicted to occupy a site similar to compound 1 or 2, it would strengthen their findings and decrease the concern (comment 4) for the inhibitors’ affecting crystal structure.

11. In most pre-clinical studies, mouse models are used. Can the authors speculate as to why they were unable to crystallize mouse HSD17B13, and test whether compounds 1 and 2 have an inhibitory effect on mHSD enzymatic activity?

Minor comments:

1. Page 8 line 3, wrong figure number.

2. Page 10 line 3, reference 11 seems to be incorrect. Did they authors mean to cite ref 23?

3. Page 13, there are multiple places that the species of HSD17B13 was not indicated. Please check throughout the paper to make sure the species are correctly indicated.

4. Figure 6B – is the inhibited vs. non-inhibited the ideal way to show the NAD⁺ preference over NADP⁺? Why not compare the two cofactors directly head-to-head?

5. Figure 6B – typo in the y-axis label

6. Page 26, please check the cell numbers are correct.

7. Supplemental Figure 6 legend – “without/without” (line 3) likely should be “with/without”.

Reviewer #2 (Remarks to the Author):

The authors describe the first atomic structures of HSD17B13, a protein of great interest for the scientific community, especially for drug discovery. The structures give insights on the mechanism of lipid droplet association which for itself is already unexpected and changes the view on the interaction of the protein with these organelles. In addition to this the authors also describe the apo structure plus two different HSD17B13-inhibitor co-structures with very different binding modes. Those structures are an excellent starting point for future structure-based drug discovery approaches, and they help to shed some light on the species differences observed for this protein. Those structures are complemented with a little bit of functional data to understand the structure-activity relationship of those compounds.

Taken together the data described in this manuscript is definitely valuable for the scientific community but could benefit from a deeper biochemical analysis of some of the structure derived hypotheses. Especially the presence of a detergent or lipid molecule in the substrate binding pocket should be studied and discussed in more detail.

Major points:

- The presence of a lipid/detergent molecule in the substrate binding pocket of HSD17B13 is an extremely important finding and it should be highlighted in more detail. It should be mentioned in the abstract and definitely in the legend of Figure 5. Some more points on this are also missing in the discussion: was any electron density for detergent molecules observed in the dog HSD17B13 structures? Is it possible to increase or decrease the amount of detergent molecules by changing the crystallization conditions? Biochemical data on the effect of lipids/detergents on the activity of the enzyme could help to better understand if this finding is physiologically relevant, or an artifact from purification. Does the detergent have an effect on thermal stability?
- The authors might have missed the new publication in JMedChem on a HSD17B13 inhibitor, which nicely fits together with their data, as some of the results are similar. It should be mentioned and commented on in the discussion.

Minor points:

- The title of the manuscript should be changed to something more descriptive
- The authors observe different conformations of the N-terminal lipid droplet anchoring helix in the human HSD17B13 structure – could this be due to crystallization contacts/packing artifacts? If they see

the more tilted conformation, is it present in both molecules of the dimer? Is it dependent on the presence of detergent/lipid?

- Line 140: should be "Figure 2"

- Line 193: should be "of a negatively charged"

- Line 257: should be "the hydroxyl of the"

- Line 269: Numbering of mutants is wrong and A is missing in "T205A"

- Line 286: should be "from the alkyl"

- Line 396: the amino acids are wrong – residues 266 are Phe and Ser and residues 281 Leu and Ile

- Line 676: should be "HSD17B13"

- Line 837: Numbering of amino acids is wrong!

- Figure legend 5 must include the description of the detergent molecule

- Figure 5A – details are hard to see

Below are point-by-point responses to the reviewers' comments.

Reviewer #1

1. Identification of inhibitors: Although the inhibitors Compound 1 and 2 are not the main focus of the manuscript, the authors should provide information on the compound library, the methodological details of how the inhibitors were selected, as well as the direct results of the luciferase assays testing them. More importantly, since the inhibitor screening assay appears to be based on purified protein, it is critical to at least verify whether the selected inhibitors indeed inhibit HSD17B13 enzymatic activity in mammalian cells.

We added in the first paragraph of the results (P5) a description of the high throughput screening that identified compounds 1 and 2. We also added a new supplementary figure to give direct results of the biochemical luciferase assay and mammalian cell assay testing these two compounds. Compound 1 is active in the cell assay, and compound 2 is inactive. We mention that it is not surprising that compound 2 is inactive in cell assay, considering it has two highly polar chemical groups (sulfonamide and sulfone) and low permeability.

2. Both inhibitor compounds were selected from the inhibitor screening using purified human HSD17B13 (hHSD) protein, and compound 1 is the stronger binder to hHSD and reported to be a stronger inhibitor. In contrast, compound 1 does not bind dog HSD17B13 (dHSD) in the TSA assay and its inhibitory capacity for dHSD is not reported. It is therefore unclear why hHSD could not crystallize with compound 1 while dHSD could. Can the authors explain this apparent paradox? They should also report whether the compounds are indeed inhibitors of dHSD. Since the crystal structure of hHSD with compound 2 was a success, why not using compound 2 for the later thermal stability testing?

We added on page 17: " Inability to detect stabilization of dog HSD13B13 by compound 1 in TSA does not mean compound 1 does not bind dog HSD17B13. In our experience TSA is insensitive to detect weak binders ($> 10 \mu\text{M } K_d$) for some proteins.". We added on page 16 "Because of the low enzymatic activities of the dog wild-type HSD17B13 (Figure 6A) and HSD17B11 with β -estradiol, it was difficult to measure compound inhibition of these proteins in a biochemical assay and we had to look to direct binding methods to compare compounds interaction with different proteins." We did include compound 2 in our TSA experiments (Figure 6D). Increasing protein stability does increase the successful rates of protein crystallization, but in our experience crystallizability of a protein does not solely depend on its stability. Addressing the crystallibility of human and dog HSD17B13, in P7 we added "On the other hand, we could not generate compound 2 bound dog wild-type HSD17B13 crystals by soaking or crystallization, and instead relied on co-crystallization with human HSD17B13." In the same paragraph we wrote "Based on findings from dog HSD17B13 crystals, we changed four surface residues distant from the active site of human HSD17B13 to those of dog protein at the dog HSD17B13 crystal interface (Q60K, I62R, R71H and E161K, human mutant) to improve resolution of the human HSD17B13 crystals, and obtained useful crystals of human HSD17B13 in complex with NAD^+ and compound 2".

3. Please provide potential explanations for human HSD17B13 forming four dimers with compound 2. Is it WT hHSD or mutant hHSD (p. 7, line 2 and Table 1)? It is unclear from the preceding paragraph.

We added in P7 "mutant" for the human HSD17B13 that crystallized with compound 2. We also added "One of the possible reasons that there are four human HSD17B13 dimers in the crystal is that the ligand binding sites contain are lipids molecules (see below) which are heterogeneous in nature, making these complexes not entirely identical structurally. "

4. Stemming from comments 2 and 3 – hHSD was crystalized with the weaker inhibitor (compound 2) in 4-dimer configuration; dHSD was crystalized with compound 1 (which doesn't bind it well) in a single dimer with only one inhibitor. This raises a concern that the compounds themselves affected, and may have altered, the crystal structure and hence, that the findings are not truly reflective of the “true” membrane-bound structure in vivo. The authors should discuss extensively why they think compounds 1 and 2 are necessary for crystallization, and the possibility of this generating a major artefact.

We added in the discussion a new paragraph (P20) on why we think compounds are necessary to crystallize these proteins. We stated that although the structure of the ligand binding site and the membrane domain of HSD17B13 depends on inhibitors, “The apo HSD17B13 has essentially the same structure in different crystals, thus it may closely resemble the protein on the surface of the lipid-droplet absence of inhibitors. The ligand binding site and the membrane anchoring domain may transiently sample other conformations due to their dynamic nature and fluidity of the membrane, allowing the lipid molecules in the lipid-droplets access the active site of HSD17B13. We also added: “One of the possible reasons that there are four human HSD17B13 dimers in the crystal is that ~~at~~ the ligand binding sites contain lipids molecules (see below) which are heterogeneous in nature, making these complexes not – structurally identical.”

5. Enzymatic activity of native HSD: Page 11 line 2-5, “When a steroid molecule was docked in the active site of the apo HSD17B13 by homology modeling (Figure 3B), the molecule would clash with the C-terminal peptide and loop P218-239. In fact, repeated soaking with substrate β -estradiol failed to result in ligand binding.” Which species of HSD17B13 was used? How do the authors reconcile this finding of blocked substrate site with the fact that they, and others, were still able to demonstrate enzymatic activity against candidate substrates (even if these may not be the physiological substrate)? Does that support the authors' interpretation of their findings to suggest a scaffolding, rather than enzymatic, role for HSD17B13?

We added “dog and human” HSD17B13 for our molecule docking, These two proteins are highly homologous. In Discussion section, P20, we added endogenous bioactive ligands may “transiently but much less efficiently access” the active site of HSD17B13 “and get turned over”. The “observed low basal enzymatic activity of the wild-type human HSD17B13” supports our findings suggesting a scaffolding role of human HSD17B13.

6. The authors identify the c-terminal helix-turn-helix domain as a second component for anchoring HSD17B13 to the LD, together with the n-terminal helix. However, Ma et al (reference 11) previously showed that despite their instability, P260S, P274del and exon 6 skipping mutants (all expected to impact the 260-286 domain) do localize to the surface of lipid droplets in mammalian cells. Can the authors reconcile this with their structure predictions?

At the end of the first paragraph of the Discussion, P19, we added “Although the N-terminal peptide 1-28 is the main membrane anchoring element of HSD17B13, the amphipathic helix-turn-helix 260-286 is also important for the stability of the protein on the surface of lipid-droplets. Disruption of the helix-turn-helix likely result in protein misfolding/instability and low protein level in cells, as is in the cases of the P260, P274del and exon 6 skipping variants of HSD17B13.”

7. In Supplemental Figure 6, the ligand to protein ratio was 10:1 and the NAD to protein ratio was 50:1, which are far beyond physiological levels. Can the authors explain why they used an inhibitor concentration (100 μ M) which is markedly higher than the IC_{50} (IC_{50} ranging from 0.18 to 4.55 μ M), and why they used 500 μ M of the cofactor NAD?

We added in P17 “Given that in theory and in practice the Tagg continues to increase with increasing concentration of ligands far above their IC₅₀s, we used high cofactor and inhibitor concentration in the TSA experiments to maximize the ligand binding signal.”

8. In Supplemental Figure 6, the authors show a correlation between Δ Tagg and IC₅₀ of multiple inhibitors, all used with at the same concentration. Isn't that effectively a surrogate for a dose response curve with a single inhibitor? We suggest the authors show a dose response curve instead.

In that figure the correlation of Δ Tagg and IC₅₀ of multiple inhibitors calibrates the Δ Tagg of a inhibitor to its IC₅₀ and is more than a dose response curve with a single inhibitor. In P17 we added “This calibration method of using multiple compounds is a better way of correlate the Δ Tagg and IC₅₀ values than dose response of a single ligand, because the K_D derived from TSA may vary with ligand concentration⁵¹”.

9. Su et al recently (reference 57) demonstrated an impact of S33 phosphorylation on liver phenotype. This serine is located just beyond the n-terminal helix that is anchored in the LD monolayer. Even if not tested experimentally, can the authors model the effect of this phosphorylation on their predicted structure?

We added P21 description of modeling of S33A and pS33. S33 is above the membrane surface and exposed to aqueous solvent, making it possible to interact with ATGL. S33A mutation increases the hydrophobicity of the residue, and phosphorylation adds a bulky charged group.

10. Thamm et al (J Med Chem, 2023) recently published a selective HSD17B13 inhibitor (BI-3231). Can the authors model its docking? If they show that BI-3231 is predicted to occupy a site similar to compound 1 or 2, it would strengthen their findings and decrease the concern (comment 4) for the inhibitors' affecting crystal structure.

We added the citation in P14. We docked BI-3231 into compound 1 binding site of HSD17B13, and added a paragraph and a Supplementary figure. In the docking model BI-3231 binds similarly to compound 1.

11. In most pre-clinical studies, mouse models are used. Can the authors speculate as to why they were unable to crystallize mouse HSD17B13, and test whether compounds 1 and 2 have an inhibitory effect on mHSD enzymatic activity?

Our purified mouse HSD17B13 has very low enzymatic activity using β -estradiol as substrate. Therefore we can't measure the inhibition of mHSD17B13 by compound 1 or 2. In the discussion we mentioned that the active site of mHSD17B13 has residues that are different from those of hHSD17B13, and these residues were proven in our mutagenesis studies to be important for inhibitor binding and enzymatic activities. Crystallization of a protein, especially a membrane protein, can be challenging and sometimes defy explanation.

Minor comments:

1. Page 8 line 3, wrong figure number.

Changed to figure 2.

2. Page 10 line 3, reference 11 seems to be incorrect. Did they authors mean to cite ref 23?

Changed to ref. 23.

3. Page 13, there are multiple places that the species of HSD17B13 was not indicated. Please check throughout the paper to make sure the species are correctly indicated.

We added species indications throughout the manuscript.

4. Figure 6B – is the inhibited vs. non-inhibited the ideal way to show the NAD⁺ preference over NADP⁺? Why not compare the two cofactors directly head-to-head?

In P16 we added “the background signal of cofactor turnover of” inhibited reactions. The assay we used detected cofactor turnover, which occurs at low level even without enzyme (oxidization by O₂ in the air), which needs to be subtracted with inhibited reaction.

5. Figure 6B – typo in the y-axis label

Corrected.

6. Page 26, please check the cell numbers are correct.

The Epxi293 cell number is corrected to 2.5×10^6 cells ml⁻¹.

7. Supplemental Figure 6 legend – “without/without” (line 3) likely should be “with/without”.

Corrected.

Reviewer #2 (Remarks to the Author):

The authors describe the first atomic structures of HSD17B13, a protein of great interest for the scientific community, especially for drug discovery. The structures give insights on the mechanism of lipid droplet association which for itself is already unexpected and changes the view on the interaction of the protein with these organelles. In addition to this the authors also describe the apo structure plus two different HSD17B13-inhibitor co-structures with very different binding modes. Those structures are an excellent starting point for future structure-based drug discovery approaches, and they help to shed some light on the species differences observed for this protein. Those structures are complemented with a little bit of functional data to understand the structure-activity relationship of those compounds.

Taken together the data described in this manuscript is definitely valuable for the scientific community but could benefit from a deeper biochemical analysis of some of the structure derived hypotheses. Especially the presence of a detergent or lipid molecule in the substrate binding pocket should be studied and discussed in more detail.

Major points:

- The presence of a lipid/detergent molecule in the substrate binding pocket of HSD17B13 is an extremely important finding and it should be highlighted in more detail. It should be mentioned in the abstract and definitely in the legend of Figure 5. Some more points on this are also missing in the discussion: was any electron density for detergent molecules observed in the dog HSD17B13 structures? Is it possible to

increase or decrease the amount of detergent molecules by changing the crystallization conditions? Biochemical data on the effect of lipids/detergents on the activity of the enzyme could help to better understand if this finding is physiologically relevant, or an artifact from purification. Does the detergent have an effect on thermal stability?

We added in the abstract that lipid/detergent molecules were observed in human HSD17B13 structures. In legend of Figure 5 we added “Solubilized lipids/detergent molecules are also observed in the ligand binding site (orange carbon, stick model).” In P6, Structure determination of HSD17B13, we did write that we screened different detergents for crystallization and added that C12E8 was identified as the suitable detergent “that provided HSD17B13 with the largest stability needed” for crystallization. We added a new paragraph in the Discussion on the relevance of the observed bound lipid/detergent molecules. We explained that these these molecules were visible in human HSD17B13 crystals likely because they were ordered by interactions with both the protein and with bound compound 2. They were in dog HSD17B13 samples, but were disordered in the crystals because we could not observe compound 2 with dog HSD17B13 because of its weak binding affinity.

- The authors might have missed the new publication in JMedChem on a HSD17B13 inhibitor, which nicely fits together with their data, as some of the results are similar. It should be mentioned and commented on in the discussion.

We added a paragraph discussing the new JMedChem paper. We added that we docked the most potent inhibitor mentioned in that paper, BI-3231, in the active site of our structure and the ligand can form similar interactions as our compounds.

Minor points:

- The title of the manuscript should be changed to something more descriptive

We change the title of the manuscript to “Structural basis of lipid-droplet localization of 17-beta-hydroxysteroid dehydrogenase 13”

- The authors observe different conformations of the N-terminal lipid droplet anchoring helix in the human HSD17B13 structure – could this be due to crystallization contacts/packing artifacts? If they see the more tilted conformation, is it present in both molecules of the dimer? Is it dependent on the presence of detergent/lipid?

In P15 we added that the N-terminal helices adopt “variable” conformations “that are all different from” those of the apo dog HSD17B13. We added “The variability of membrane domains of human HSD17B13 arise from heterogeneity of bound lipids/detergent molecules with which they interact”.

- Line 140: should be “Figure 2”

Corrected.

- Line 193: should be “of a negatively charged”

corrected

- Line 257: should be “the hydroxyl of the”

corrected

- Line 269: Numbering of mutants is wrong and A is missing in “T205A”

corrected

- Line 286: should be “from the alkyl”

corrected

- Line 396: the amino acids are wrong – residues 266 are Phe and Ser and residues 281 Leu and Ile

corrected

- Line 676: should be “HSD17B13”

Corrected

- Line 837: Numbering of amino acids is wrong!

Corrected

- Figure legend 5 must include the description of the detergent molecule

Included

- Figure 5A – details are hard to see

We generated a new Figure 5A emphasizing ligand binding site and N-terminal helix.

REVIEWERS' COMMENTS

Reviewer #1 (Remarks to the Author):

The authors have adequately addressed most of our comments. We suggest the following minor changes (numbered according to the numbering of our original review):

1. Please provide technical information on the compound library used for screening (i.e. manufacturer and catalog number). If proprietary to a company and not available, please state so.
3. It is still unclear at multiple sections in the manuscript (i.e. Table 1, figure 5 legend , results paragraph titled "Complex structure of compound 2 with human HSD17B13" and elsewhere) when the authors use "human HSD17B13" if they mean the WT human or mutant human protein. We suggest labeling all occurrences of "human HSD17B13" with "mutant" or "WT" to help clarify for readers.

Reviewer #2 (Remarks to the Author):

Thank you for your revisions. My concerns are now addressed and I recommend the manuscript for publication.

1. Please provide technical information on the compound library used for screening (i.e. manufacturer and catalog number). If proprietary to a company and not available, please state so.

We added that it is a “Pfizer proprietary” compounds library.

2. It is still unclear at multiple sections in the manuscript (i.e. Table 1, figure 5 legend, results paragraph titled “Complex structure of compound 2 with human HSD17B13” and elsewhere) when the authors use “human HSD17B13” if they mean the WT human or mutant human protein. We suggest labeling all occurrences of “human HSD17B13” with “mutant” or “WT” to help clarify for readers.

We added “mutant” in Table 1, figure 5 legend, results paragraph wherever human HSD17B13 crystal structures were mentioned.